# Selecting the Best Performing Modified Asphalt Based on Rheological Properties and Microscopic Analysis of RPP/SBS Modified Asphalt

**DOI:** 10.3390/ma15238616

**Published:** 2022-12-02

**Authors:** Lei Guo, Wenyuan Xu, Yang Zhang, Weishuai Ji, Suxin Wu

**Affiliations:** 1School of Civil Engineering, Northeast Forestry University, Harbin 150040, China; 2Heilongjiang Communications Investment Group Co., Ltd., Harbin 150001, China

**Keywords:** RPP, rheological properties, micro analysis, analytic hierarchy process, optimum dosage

## Abstract

As an asphalt modifier, waste polypropylene (RPP) can not only optimize the performance of asphalt but also greatly alleviate the problem of waste plastic treatment, effectively reducing environmental pollution and resource waste. In order to evaluate the influence of RPP and styrene butadiene styrene (SBS) on asphalt performance, the application of RPP in modified asphalt pavement has been expanded. In this study, a dynamic shear rheometer (DSR), bending beam rheometer (BBR) and other instruments were used to evaluate the rheological properties of composite-modified asphalt. Fourier infrared spectroscopy (FTIR) and fluorescence microscopy (FM) was employed to conduct a microscopic analysis of the modified asphalt, and the layer analysis method was adopted to determine the optimal RPP content. The test results show that the rheological properties of asphalt are significantly improved by the composite modification of RPP and SBS. In addition, the cross-linking between polymer and asphalt is further enhanced by the composite addition of RPP and SBS. The comprehensive performance of modified asphalt is optimized at the RPP content of 2%, which is suitable for applications in the cold temperate zone. The RPP/SBS composite-modified asphalt is able to improve the utilization rate of RPP and has good environmental and economic benefits, thus exhibiting excellent comprehensive performance. However, the optimal asphalt content in the mixture was not investigated, and the economic benefits brought by the utilization of RPP were not evaluated and require further study.

## 1. Introduction

Asphalt is a viscoelastic material that has gradually become one of the most widely used road pavement materials in China [1,2] due to advantages including low noise and surface evenness. However, in recent years, with the increase of heavy vehicles and the complexity of paved environments, ordinary matrix asphalt is sensitive to the subsequent changes in external temperature. Irreversible elastic deformation and viscous deformation will consequently occur under the action of external forces. In particular, at high temperatures, the elastic deformation of the pavement reduces and viscous deformation increases, which will cause serious rutting problems under repeated traffic loads [1]. In addition, asphalt pavement can easily crack when it is used at low temperatures and is thus unable to meet the requirements of the majority of environments. The construction, maintenance, repair, restoration and reconstruction of new roads and the maintenance of aging pavements require a large number of materials and unproductive energy consumption, which has a huge cost demand for the economy [3]. In order to improve the road performance of asphalt and save costs, scholars have focused more attention on the research on modified asphalt in recent years. Recycling waste plastics to change asphalt not only optimizes the road performance of asphalt but also saves costs and protects the environment [4,5].

Due to their low cost and high durability, plastics are widely used in various fields of life [6]. Most plastic wastes are divided into four categories: polyester; polyolefin; polyvinyl chloride (PVC); and polystyrene (PS). Polyolefins, such as polyethylene (PE) and polypropylene (PP), have an annual output of approximately 218 million tons, accounting for 57% of the plastic content of municipal solid waste [7]. Among them, the low cost, excellent high-temperature performance, and chemical corrosion resistance of PP plastic have made it a widely popular material. At the same time, RPP plastic requires extensive land resources due to its non-degradability and causes serious environmental pollution [8]. Therefore, the recycling and management of RPP plastic are urgent issues that need addressing [9]. At present, the most common recovery methods are mechanical and chemical recovery [10]. Recently, researchers have further expanded the application scope of recycled RPP, and a more economical and practical usage is in infrastructure construction (including modified asphalt) [8,11,12]. Moreover, many scholars have studied this field for decades [13]. PP, a high molecular polymer, is a thermoplastic plastic with stable physical and chemical properties that can be used to modify base asphalt. RPP-modified asphalt can effectively change the structure of raw asphalt collage and form new collage structures, thus improving the high-temperature resistance, moisture susceptibility, and other properties of the asphalt mixture. This consequently improves the pavement quality, saves maintenance costs, and extends the service life of the asphalt pavement [14,15]. 

Rubber modification can improve the low-temperature toughness of PP. In the past two decades, researchers have attempted to add elastomer or rubber to RPP, with materials including ethylene-propylene copolymer, ethylene propylene diene rubber, styrene butane styrene, etc., achieving some improvements [16]. Zhao et al. evaluated the performance of polypropylene (PP, 80–85%) and polyethylene (PE, 15–20%) copolymers as modifiers. The authors found that 6% of RPP/PE copolymers enhanced the rutting resistance in the use range of 50–80 °C, improved the fatigue performance, and had the least impact on low-temperature performance [17]. However, the low-temperature performance of modified asphalt decreased slightly. Moatafa et al. compared the fatigue resistance of RPP/butadiene styrene rubber (SBR) composite-modified asphalt mixture with that of the SBS-modified asphalt mixture and determined that at the 5% content of the (0.3 PP + 0.7 SBR) blend, the fatigue resistance of the composite modified asphalt mixture is more than 50% higher than that of the 5% SBS modified asphalt mixture [18]. Cheng et al. mixed two types of waste (PP and SBR) into the base asphalt, respectively. The authors tested and compared the performance of asphalt and the mixtures to conclude that the high-temperature performance of PP/SBR composite-modified asphalt was improved, and the low-temperature performance was also slightly improved [4]. However, the previous testing showed that the addition of SBR did not significantly improve the low-temperature performance of PP-modified asphalt, with a significant gap compared with the SBS-modified asphalt. The addition of SBR has also been observed to slightly affect the high-temperature performance of modified asphalt [4,18]. Therefore, it is necessary to determine a material that can significantly improve the low-temperature crack resistance of PP-modified asphalt without affecting the other properties. SBS-modified asphalt is widely used in high-grade asphalt pavement surface courses and has excellent high and low-temperature performances, as well as strong fatigue resistance [19]. SBS is a rubber plastic material [20], and the polystyrene inside allows for good high-temperature performance, while the polybutadiene facilitates high flexibility at low temperatures [21]. In addition, since SBS does not contain C = C, the SBS-modified asphalt mixture also exhibits a good low-temperature cracking resistance [22]. Therefore, SBS is often combined with other modifiers and can be employed as a material to improve the low-temperature performance of PP-modified asphalt. 

Previous explorations of asphalt or modifier dosages include the application of multiple expression programming (MEP) to develop empirical prediction models for Marshall parameters in order to obtain the optimal asphalt content and asphalt pavement-related parameters (Awan et al.) [23]. The results reveal that the developed models surpass their predecessors in terms of the prediction and generalization of re-output parameters. However, such models are prediction models. Based on the existing experimental research, this study must determine a method to optimize the optimal amount of RPP in modified asphalt with quantitative indicators. Analytic Hierarchy Process (AHP) is widely used in system engineering. It essentially establishes a judgment matrix by comparing two factors and takes the ranking weight value based on the judgment matrix. It has obvious systematic and comprehensive characteristics [24] and is a suitable evaluation method for this study

The purpose of this paper is to make use of the advantages of RPP that can significantly enhance the high-temperature rutting resistance and temperature sensitivity of asphalt, and SBS that can enhance the low-temperature toughness of asphalt to produce RPP/SBS composite modified asphalt with better high- and low-temperature performances. Three index tests, the dynamic shear rheology test and the bending beam, creep stiffness test, were performed to analyze the improvement of RPP and SBS at the high and low temperatures of base asphalt under different scenarios. In particular, the modification and compatibility mechanisms were analyzed at the micro level through infrared spectrum scanning and fluorescence microscope tests. The comprehensive performance index of modified asphalt with different RPP contents under typical traffic environment conditions in Northeast China was calculated by combining various properties of modified asphalt with an analytic hierarchy process, and the modifier content was optimized to determine the optimal RPP content with quantitative indicators. This study promotes the application of waste plastics in road engineering and environmental protection and can help to reduce waste and save costs. The specific research method of this study is listed in Figure 1. 

## 2. Materials and Methods

### 2.1. Test Materials

#### 2.1.1. SBS Modified Asphalt

SBS-modified asphalt is prepared by mixing SBSYH-792E modifier (produced by Sinopec, see Table 1 for basic performance parameter) and SK-90 base asphalt (Hohhot, Inner Mongolia, China) in the laboratory (Table 2), with an SBS modifier content of 4%. According to previous experimental studies, when SBS and polyolefin modifiers are combined, the comprehensive road performance of modified asphalt is optimized at the SBS content of 4% [25]. Figure 2 depicts the proposed modifier. 

#### 2.1.2. Waste Polypropylene (RPP)

A recycled polypropylene woven bag (polypropylene used as raw material) was selected as the modifier, and its basic physical properties are shown in Table 3. The bag was washed with clean water, placed into an oven to dry the surface moisture, and subsequently cut into 1 cm (side length) pieces to be used as an asphalt modifier [3]. Figure 3 presents the waste PP modifier. 

### 2.2. RPP/SBS Modified Asphalt Preparation

In order to fully mix the modifier with the base asphalt, the modified asphalt was prepared by melting and mechanical blending. Based on the optimal preparation process of RPP-modified asphalt described in previous research [26], this study heated the base asphalt at 135 °C to a molten state and placed it on a 165 °C hot plate for the addition of the SBSYH-792E modifier. The mixture was then stirred at a speed of 400 r/min for 30 min until there were no obvious particles on the surface of the blend. Following this, 1%, 2%, 3%, 4% and 5% RPP were added, respectively, and the mixture was placed on a heating plate, heated to 170 °C, stirred at 500 r/min for 30 min, and blended at a rotor speed of 5000 r/min. This was followed by shearing at 175 °C for 90 min and swelling at 160 °C for 45 min to obtain the RPP/SBS composite modified asphalt. Figure 4 presents the specific preparation process. 

### 2.3. Test Methods

#### 2.3.1. Routine Performance Test of Asphalt

The softening point (R&B), penetration (25 °C), and ductility (5 °C) of the RPP/SBS modified asphalt were tested in accordance with the test standards in the *Standard Test Methods of Bitumen and Bituminous Mixtures for Highway Engineering (JTGE20-2011)* [27].

#### 2.3.2. Dynamic Shear Rheometer Test

The dynamic shear rheometer (Anton Paar MCR 302 DSR instrument) (Figure 5) was used for the temperature scanning experiment to characterize the rheological properties of the RPP/SBS modified asphalt. During the experiments, a continuous sinusoidal alternating load was applied, and the strain control mode was adopted [28]. The test metal plate diameter was 25 mm, and the asphalt sample thickness was 1.0 mm. Moreover, the test temperature, strain level and rotation frequency were set as 30–85 °C, 1%, and 10 rad/s, respectively.

#### 2.3.3. Bending Beam Creep Test at Low Temperatures

The TE-BBR-F bending beam rheometer (CANNON, Melville, NY, USA) (Figure 6) was employed for the bending creep tests of RPP/SBS modified asphalt at −12 °C, −18 °C and −24 °C under the loading process of 240 s. The low-temperature crack resistance of the modified asphalt was evaluated according to the stiffness modulus, creep rate and k index recorded at the 60th s [29]. 

#### 2.3.4. Infrared Spectrum Test

The modified asphalt samples were dissolved in dichloromethane to produce a 10% solution. The effects of the different RPP modifier dosages on the chemical composition and functional groups of SBS-modified asphalt were evaluated by Fourier transform infrared spectroscopy (FTIR) (Figure 7) at a 4 cm^−1^ resolution, the number of scans was 32, and the wavenumber test range was 4000 to 400 cm^−1^ [30].

#### 2.3.5. Fluorescence Microscope

The dispersion of the polymer modifier in the asphalt was observed using an upright high-resolution fluorescence microscope (FM) (Figure 8). In particular, a Nikon two-photon laser confocal superresolution microscope was employed at a magnification of 10 × 10. The observed asphalt samples were prepared by dropping hot asphalt into glass slides and pressing them into thin layers with cover glass [31]. Under fluorescence irradiation, the polymer was generally observed as green and the base asphalt as black.

## 3. Results

### 3.1. Analysis of Basic Physical Properties of RPP/SBS Modified Asphalt

The penetration (25 °C), softening point and ductility (5 °C) of modified asphalt was tested according to the relevant requirements and test methods of the *Standard Test Methods of Bitumen and Bituminous Mixtures for Highway Engineering (JTGE20-2011)*. Figure 9 depicts the influence of RPP content on the three indicators of modified asphalt. 

The penetration value of RPP/SBS composite-modified asphalt is lower than that of base asphalt and SBS-modified asphalt, while the softening point value is higher than that of SBS-modified asphalt. After adding RPP, the 5 °C ductility value of composite-modified asphalt decreases compared with that of SBS-modified asphalt, yet the reduction is small. Moreover, at the RPP content of 2%, the softening point value of the composite-modified asphalt increases by 20% compared with the SBS-modified asphalt, and the ductility value decreases by 5.01%. This indicates that the high-temperature performance of PP/SBS composite-modified asphalt prepared with an appropriate amount of RPP is significantly improved compared with SBS-modified asphalt, and the low-temperature drop is very small. 

### 3.2. Rheological Properties of RPP/SBS Composite Modified Asphalt

#### 3.2.1. Temperature Scanning Test and Analysis

DSR testing was performed to determine the complex shear modulus G*, phase angle sin δ, and rutting factor G*/sin δ. The complex shear modulus G* is a measure of the total resistance of materials during repeated shear deformation. The larger the values of G*, the greater the stiffness of asphalt, the better its high-temperature stability and the stronger its ability to resist flow deformation. The larger the value of G*/sin δ, the better the rutting resistance of RPP/SBS composite-modified asphalt at high temperatures [32]. Based on the statistics of the temperature scanning results of modified asphalt at 64–82 °C (Table 4), the high-temperature performance of asphalt is analyzed by plotting Figure 10 and Figure 11 and Table 4 and Table 5 with temperature and RPP content as the independent variables.

The composite shear modulus G* represents the ability of the sample to resist deformation during repeated shearing. It is the ratio of the maximum shear stress to the maximum shear strain. Figure 10 reveals that, within the test range of 64–82 °C, the G* values of the matrix asphalt, SBS modified asphalt and RPP/SBS composite modified asphalt all decrease with the increasing temperature. This is attributed to the reduction in the force and the cross-linking relationship between asphalt molecules as the temperature increases, as well as the lower asphalt elasticity. Thus, the viscosity component increases while the anti-deformation ability of asphalt and the G* value decreases. At 76–82 °C, the change in the asphalt G* value is generally minimal, which indicates that asphalt at high temperatures lower than 76 °C is more sensitive to temperature. Within the range of 64–76 °C, temperature factors play a decisive role in the composite shear modulus of asphalt.

By comparing the data in the chart under the same temperature, the G* values of the modified asphalt types are in the order of RPP/SBS modified asphalt > SBS modified asphalt > base asphalt. With the addition of RPP, the G* value of modified asphalt gradually increases at 64 °C. At the RPP content of 5%, the G* value of composite-modified asphalt increases by 118% compared with SBS-modified asphalt, indicating that the addition of RPP changes the viscoelastic composition of asphalt. In particular, RPP/SBS modified asphalt exhibits a greater number of elastic components at the same temperature, increases the cohesion of the asphalt binder, and thus enhances the resistance of the modified asphalt to deformation at high temperatures.

According to Superpave regulations, when G*/sin δ = 1, the corresponding temperature is the failure temperature of the asphalt. Figure 11 demonstrates that after adding the RPP, the rutting factor of RPP/SBS composite-modified asphalt exhibits a significant increase, and the failure temperature of SBS-modified asphalt has been raised to above 76 °C. This indicates that asphalt has a greater resistance to permanent deformation under high-temperature conditions, and as the temperature increases, G*/sin δ declines sharply and subsequently stabilizes. Under the same temperature, the G*/sin of composite-modified asphalt δ gradually increases with the RPP content, and the maximum value is approximately twice that of the SBS-modified asphalt. Under different dosages and temperatures, the change slope of the rutting factor is similar to that of the composite shear modulus. This highlights the strong influence of G* on the changing trend of the rutting factor, and the addition of RPP can effectively improve the high-temperature rutting resistance of asphalt. In particular, the higher the dosage, the better the high-temperature performance of modified asphalt. 

#### 3.2.2. Multi-Stress Creep Test Analysis and Results

In order to better evaluate the high-temperature performance of RPP/SBS composite-modified asphalt, MSCR tests were conducted based on the DSR. The high-temperature performance of asphalt was evaluated by recording the delayed elastic recovery deformation and irrecoverable deformation of road asphalt under the action of force. Applying stress to the asphalt sample will cause deformation. After removing the stress, part of the deformation can be observed in the form of delayed recovery, and the irrecoverable deformation will be accumulated to the next cycle load. Under the periodic action of unloading, the driving load of the pavement can be precisely simulated, and the high-temperature anti-rutting performance can be analyzed more accurately with temperature scanning [33]. The average strain recovery rate (R) and irrecoverable creep compliance ($J_{nr}$) are commonly used to evaluate the test results. The average strain recovery rate reflects the recovery ability of asphalt samples after elastic deformation. The larger the value of R, the higher the elasticity of the asphalt binder and the better the high-temperature deformation resistance. The irrecoverable creep compliance ($J_{nr}$) reflects the ability of asphalt mastic to resist permanent deformation. The greater its value, the weaker its rutting resistance at high temperatures. R and $J_{nr}$ are calculated following Formulas (1) and (2):(1)$$R=0.1\sum_{i=1}^{10}\frac{\sigma_{ip}-\sigma_{inr}}{\sigma_{ip}-\sigma_{io}},$$
(2)$$J_{nr}=0.1\sum_{i=1}^{10}\frac{\sigma_{inr}-\sigma_{io}}{\tau },$$
where $\sigma_{p}$ (%) is the peak strain of each cycle; $\sigma_{o}$ (%) is the contingency for each cycle; $\sigma_{nr}$ (%) is the residual strain of each cycle, and *τ* (kPa) is the loading stress.

Figure 12 and Figure 13 present the values of R and $J_{nr}$ calculated under two stress states (0.1 kPa and 3.2 kPa), respectively. Under the two stress conditions, as the temperature increases, the R and J_nr_ values of the seven asphalt types are observed to decrease continuously. In addition, the R and $J_{nr}$ values under the stress condition of 3.2 kPa change more than those under the 0.1 kPa stress condition. This indicates that high temperature and stress will reduce the elastic recovery ability and deformation resistance of asphalt. In general, the values of R and $J_{nr}$ are observed to increase, revealing that the sensitivity of asphalt to temperature increases with temperature. This is due to the stronger fluidity of asphalt at higher temperatures, which makes it difficult for it to recover in the creep recovery stage, thus reducing the average strain recovery rate with the increasing temperature. 

Figure 12 reveals that under the two stress conditions, the R-value of the modified asphalt increases with the RPP content while the *J_nr_* value decreases. This indicates that the addition of RPP changes the viscoelastic properties of the asphalt, enhancing the elastic recovery performance of the modified asphalt and reducing the viscous deformation. Moreover, at the RPP content of 1% and 2%, the viscoelastic properties are observed to improve significantly, while at the RPP content of 3–5%, the viscoelastic properties slightly improve. The test results are consistent with the temperature scanning observations.

### 3.3. Low-Temperature Cracking Performance of RPP/SBS Composite Modified Asphalt

BBR tests can characterize the influence of RPP content on the low-temperature creep performance of asphalt. The creep properties of modified asphalt with different RPP contents of −12 °C, −18 °C and −24 °C were tested by the BBR, with a stiffness modulus S and creep rate m 60 s. Stiffness modulus S refers to the low-temperature cracking resistance of asphalt. The smaller the value of S, the better the low-temperature performance of asphalt. Creep rate m represents the stress relaxation ability and sensitivity of asphalt binder stiffness with time. The larger the creep rate m, the stronger the cracking resistance of asphalt at low temperatures. Figure 14 and Figure 15 present the change trends of S and m for RPP/SBS composite-modified asphalt with RPP content, respectively.

Figure 14 and Figure 15 reveal that the stiffness modulus of the matrix asphalt, SBS-modified asphalt and RPP/SBS composite-modified asphalt increases as the temperature decreases. However, the opposite is observed for the creep rate; namely, when the temperature decreases, the asphalt brittleness increases, while the temperature tensile stress and resistance to the low-temperature cracking of materials both decrease.

At the three temperatures, the stiffness modulus S increases slowly with the addition of RPP, while the creep rate m decreases slowly. Compared with the SBS-modified asphalt, when the content of RPP is less than 5%, the S (m) value of the RPP/SBS composite-modified asphalt increases (decreases) slightly. In contrast, compared with the base asphalt, the S and m of RPP/SBS composite-modified asphalt exhibit obvious changes.

According to the Superpave regulations, S < 300 MPa and m ≥ 0.3 at the test time of 60 s. At −24 °C, the S value of the matrix asphalt exceeds 300 MPa, while m is less than 0.3 (Figure 13), which does not meet the standard. However, at −18 °C, the S value meets the standard while the m value does not, although it is close to the standard. Therefore, the low-temperature performance of the base asphalt at −18 °C cannot be clearly evaluated. 

The test results demonstrate that the low-temperature performance of the matrix asphalt is more sensitive to temperature, and the addition of RPP may affect the interaction between asphalt molecules, thus reducing the toughness of modified asphalt. However, the decrease is small. Following the addition of SBS, the low-temperature crack resistance of RPP/SBS composite-modified asphalt is obviously improved compared with the matrix asphalt. 

As the low-temperature performance of modified asphalt is affected by factors such as cracking resistance and the stress relaxation ability at low temperatures, considering only the value of S or m is a limiting factor of the analysis. Therefore, in order to simultaneously consider the low-temperature cracking resistance and stress relaxation ability of modified asphalt, the k index is determined. In particular, the k index is strongly correlated with the mixture; its calculation is quick and convenient, it has a high test accuracy, and it can accurately distinguish the low-temperature performance difference between matrix and modified asphalt [34]. The k value is determined as: (3)$$k=\frac{S\times 10^{-3}}{m},$$
where S and m are the stiffness modulus and creep rate of modified asphalt, respectively.

At the same temperature, the lower the stiffness modulus of modified asphalt, the higher the creep rate, indicating the better low-temperature performance of asphalt. Thus, the lower the k value, the better the low-temperature cracking resistance and relaxation ability of asphalt. Taking −18 °C as an example, we calculate the k values of matrix asphalt, SBS-modified asphalt and RPP/SBS composite-modified asphalt under different RPP contents (Figure 16). 

As the Superpave requires S < 300 MPa and m ≥ 0.3, the upper limit of the k value is 1. As shown in Figure 16, the k value of the base asphalt does not meet the standard at −18 °C and −24 °C, while the SBS modified asphalt meets the requirements at RPP/SBS composite modified asphalt at all temperatures. The k index also accounts for the anti-cracking performance and stress relaxation capacity of the asphalt binder at low temperatures and hence can more accurately detect changes in the low-temperature performance of asphalt under different RPP contents.

### 3.4. Study on the Modification Mechanism of RPP

#### 3.4.1. Fourier Infrared Spectrum

Figure 17 presents the FTIR spectra of modified asphalt, allowing us to compare the spectral band strength of RPP/SBS composite modified asphalt and SBS-modified asphalt. Due to the different RPP contents, the specific vibrations produced after the molecules are excited vary with the atomic groups of the modified asphalt, and characteristic absorption peaks appear in the infrared spectrum. By observing each absorption peak, we quantitatively analyzed the mechanism of the asphalt modification. 

The FTIR spectra of RPP/SBS modified asphalt exhibit strong absorption peaks at 2925 cm^−1^ and 2850 cm^−1^, where the symmetric and antisymmetric tensile vibrations of -CH- and -CH_2_- saturated hydrocarbons and their derivatives indicate that the modified asphalt contains saturated hydrocarbons and their derivatives. As the RPP content increases, the two absorption peaks are obviously enhanced, indicating that RPP content will increase the content of saturated hydrocarbons in asphalt.

A weak absorption peak appears in the wavenumber range of 2200–2400 cm^−1^, revealing that the asphalt exhibits C≡C expansion vibration in this band. The absorption peak at 2305 cm^−1^ demonstrates that the RPP addition will increase the content of C≡C in asphalt.

The vibration of asphalt within the 1600 cm^−1^ band denotes the absorption spectrum band of aromatic C = C and multi-conjugated hydrogen bound C = O stretching vibration. No obvious changes are observed in the absorption peak at the 1600 cm^−1^ band as the RPP content increases, yet it is significantly stronger than that of the SBS-modified asphalt. This indicates that RPP content will affect the change of C = C and C = O bonds but has no obvious relationship with RPP content.

The absorption peak vibrations of asphalt at 1450 cm^−1^ and 1265 cm^−1^ represent the variable angle vibration of -CH3, -CH2- and the umbrella vibration of -CH3, respectively. The variation trend is the same as that at 2925 cm^−1^, verifying that the addition of RPP will increase the content of saturated hydrocarbons in asphalt.

The absorption peaks in the 700–900 cm^−1^ band represent the out-of-plane bending of C-H bonds and the bending vibration of rings in aromatic compounds. The absorption peaks in this section indicate that the asphalt contains benzene substituents or adjacent hydrogen atomic groups. Figure 17 reveals that the absorption peak is slightly enhanced with the incorporation of RPP, demonstrating that the incorporation of RPP can slightly increase the C-H bond content in the aromatic group.

No new absorption peaks are observed in the entire functional group area of RPP/SBS composite-modified asphalt. Therefore, without the addition of other modifiers, the modification of asphalt by RPP is attributed to the physical modification, namely, the improvement in the stability of the modified asphalt system by adding morphological changes in asphalt after the review of RPP and SBS. This enhances the high-temperature performance of asphalt without the occurrence of chemical reactions.

#### 3.4.2. Fluorescence Microscope Images and Compatibility Analysis

The FM image of polymer-modified asphalt can generally be divided into two different phases: the asphalt phase and the polymer phase. However, the dominant fluorescent phase in asphalt is the aromatic phase. When the modifier absorbs aromatic hydrocarbons, the polymeric additive will brighten under fluorescence [35]. When the modifier absorbs aromatic hydrocarbons, the polymerization additive will brighten the FM images of the matrix asphalt, SBS-modified asphalt and RPP/SBS composite-modified asphalt under fluorescence (Figure 18). Figure 18b,c reveals that RPP is distributed in a spherical shape in the asphalt. Although the distribution is relatively uniform, there is no connection between them. The sphere diameter increases with the RPP content. Figure 18e,f shows the network structure following the addition of 4% SBS to the modified asphalt. As the RPP content increases, the network structure disappears and gradually agglomerates into a whole entity. This indicates that the excessive addition of RPP will cause the polymer to gather together, which may induce the storage stability problem of the modified asphalt. Moreover, since the low-temperature toughness of RPP is low, the low-temperature performance of the modified asphalt will further decline after the polymer is agglomerated. In summary, the addition of suitable amounts of RPP and SBS will form a network structure inside the composite modified asphalt to cross-link with the asphalt, which is conducive to the improvement of the asphalt performance. However, this will not be the case if too much RPP is added. 

## 4. Optimization of RPP/SBS Composite Asphalt

Common, comprehensive evaluation methods include the Delphi method, the SAW method, the analytic hierarchy process and the fuzzy comprehensive evaluation method [24]. Among them, the Analytic Hierarchy Process (AHP) is widely used and belongs to the category of system engineering methods. Due to the outstanding systematicness and hierarchy of AHP, the establishment of an AHP model can organize complex problems and quantitatively reflect the subjective judgment on the importance of multiple factors, thus avoiding the error caused by subjective judgment. Therefore, this study employs AHP to optimize the RPP content in RPP/SBS composite-modified asphalt, taking the Northeast region of China as an example.

### 4.1. Selection of Evaluation Elements Based on Asphalt Performance

When comprehensively evaluating asphalt performance, various factors such as environmental climate, traffic grade and the performance of the asphalt should be comprehensively considered. The factors affecting the service performance of asphalt pavement generally require high-temperature rutting resistance, low-temperature resistance, temperature sensitivity and the anti-aging performance of asphalt binder. Therefore, the aforementioned properties are selected as the optimal evaluation elements of asphalt performance in this study.

### 4.2. Optimization of Evaluation Indicators

Based on previous research, and considering that the climate in Northeast China exhibits large temperature differences in the four seasons and low temperatures in winter, the K index and ductility at −18 °C were adopted to evaluate the low-temperature performance of modified asphalt. Furthermore, the rutting factor G*/sin at 64 °C was used to assess the high-temperature performance of modified asphalt. The temperature sensitivity of modified asphalt was investigated by the penetration index PI, while the anti-aging performance was evaluated by the short-term aging residual penetration. PI was calculated as follows:(4)$$PI=\frac{20-500A_{\mathrm{lg}Pen}}{1+50A_{\mathrm{lg}Pen}},$$
where $A_{\mathrm{lg}Pen}$ is the regression line slope of penetration at different temperatures. The penetration tests of five types of modified asphalt with different RPP content were performed under three temperatures (15 °C, 25 °C, and 30 °C).

### 4.3. Grading Standards of Multiple Technical Indexes for Modified Asphalt

The scoring standard essentially divides the values of each index into four grades (e.g., excellent, good, fair, and pass) on the basis that each technical index meets the current Technical Specification for the Construction of Highway Asphalt Pavement. The proportions of the grades are 20%, 30%, 30% and 20%, with corresponding scores of 9, 8, 7 and 6, respectively. If a technical index does not meet the specification requirements, the asphalt cannot be used in the project. Table 6 reports the numbers for each group of modified asphalt. The values of each evaluation index are shown in Table 7, while Table 8 reports the grade score intervals, and the scores of each group of modified asphalt are listed in Table 9.

### 4.4. Weight Determination of Optimal Evaluation Elements

#### 4.4.1. Determination of Weight Based on the Comparison Matrix Method

In the analytic hierarchy process, the comparison matrix method is used to determine the weight of each element, which can clarify the fuzzy concept and determine the important order of each element. First, the four evaluation elements are arranged into 4 × 4 matrices. The matrix values are determined according to the importance of each element by pairwise comparisons of the elements. The maximum eigenvalue of the matrix and corresponding maximum eigenvector is then calculated, and a consistency check is conducted. If the consistency check is passed, the maximum eigenvector is considered to be the weight vector. Using this method, the evaluation element universe $R,\;R=\left(r_{1},\;r_{2},\;r_{3},\;r_{4}\right),$ is established, and two elements $r_{i}\;\left(i=1,\;2,\;3,\;4\right)$ and $r_{j}\;\left(j=1,\;2,\;3,\;4\right)$ are selected to compare the importance of the two elements; $r_{ij}$ represents the judgment value of the importance of element $r_{i}$ to element $r_{j}$. Table 10 reports the corresponding relationship between the importance and actual values.

Table 11 presents the established comparison matrix model. In this study, high-temperature performance is selected as the benchmark. With the exception of the high-temperature performance, the importance of each factor is calculated according to the high-temperature performance. Through the pairwise comparisons of each factor, we get:$$r_{ij}\;\left(i=1,\;2,\;3,\;4;\;j=1,\;2,\;3,\;4\right)$$
and establish the following matrix:$$R=\left(\begin{array}{cccc} r_{11} & r_{12} & r_{13} & r_{14}\\ r_{21} & r_{22} & r_{23} & r_{24}\\ r_{31} & r_{32} & r_{33} & r_{34}\\ r_{41} & r_{42} & r_{43} & r_{44} \end{array}\right).$$

The northeast region of China (2-1 and 2-2) is selected to calculate the performance weights. According to the climate partition-temperature chart of asphalt pavement in China, the high and low-temperature grades of each climate partition can be determined, and the temperature difference range of each climate partition can be estimated. Using the temperature difference range, each climate partition can be roughly divided into three grades, which are respectively denoted by reference numbers 1, 2 and 3. Previous studies indicate that the main factors affecting the aging of asphalt pavement are temperature and the ultraviolet grade [24]. Therefore, China’s annual total solar radiation distribution map divides China’s radiation intensity into three grades, which are also represented by 1, 2 and 3. The lower the label, the worse the climatic condition. Table 12 reports the climatic factor grades of two sub-regions in Northeast China.

The 2-1 and 2-2 partition judgment matrices are established as follows:$$2\text{-}1\;\mathrm{partition}:\;\left(\begin{array}{cccc} 1 & 1/1.5 & 1/1.5 & 1\\ 1.5 & 1 & 1 & 1.5\\ 1.5 & 1 & 1 & 1.5\\ 1 & 1/1.5 & 1/1.5 & 1 \end{array}\right)$$
$$2\text{-}2\;\mathrm{partition}:\;\left(\begin{array}{cccc} 1 & 1 & 1 & 1.5\\ 1 & 1 & 1 & 1.5\\ 1 & 1 & 1 & 1.5\\ 1/1.5 & 1/1.5 & 1/1.5 & 1 \end{array}\right),$$
and the eigenvectors of these matrixes are determined as:$$W_{2\text{-}1}={\left(0.2,\;0.3,\;0.3,\;0.2\right)}^{T},$$
$$W_{2\text{-}2}={\left(0.273,\;0.273,\;0.273,\;0.182\right)}^{T}.$$

#### 4.4.2. Consistency Checking of Judgment Matrix

In order to improve the reliability of the evaluation results, it is necessary to check the consistency of the judgment matrix. The criterion for the consistency of the judgment matrix is *CI* = 0 [36], and *CI* is calculated as follows:(5)$$CI=\frac{\lambda_{max}-n}{n-1},$$
where CI is the matrix consistency index; and $\lambda_{max}$ is used to judge the maximum eigenvalue of the matrix, $\lambda_{max2\text{-}1}=4$, $\lambda_{max2\text{-}2}=4$.

For both partitions, $\lambda_{max}$ = 4, and thus the CI values of the two partitions are 0. This indicates that the matrix has complete consistency and hence has high reliability. Table 13 reports the recommended performance weights in the two climate zones.

#### 4.4.3. Calculation of Asphalt Comprehensive Performance Index

The scoring basis and performance scoring results of the road performance evaluation indexes obtained in Section 4.4.1 are combined with the weights of the road performances determined in Section 4.4.2 to derive the comprehensive performance index of asphalt. The low-temperature performance of asphalt is jointly evaluated by the k index at 5 °C and −18 °C, and thus each accounts for 50% of the low-temperature performance weight. P represents the comprehensive performance index calculated according to the technical index score of the asphalt road performance weight level, p.
(6)$$P=A\omega_{A}+B\omega_{B}+C\omega_{C}+D\omega_{D}+E\omega_{E},$$
where $\omega_{A},\;\omega_{B},\;\omega_{C},\;\omega_{D},\;\mathrm{and}\;\omega_{E}$ are the rutting factor, ductility, k index, penetration index PI, and weight coefficient of the penetration ratio after aging, respectively. Table 14 reports the comprehensive performance index of asphalt under the two climatic zones.

Asphalt B (2% RPP/4% SBS composite modified asphalt) is observed to have the highest score. Compared with the SBS-modified asphalt, the scores obtained by combining the performance indicators of the two partitions increase by approximately 11.8% and 13.4%, respectively. Therefore, considering the multiple technical performance indicators, the optimal content of modified asphalt is 2% RPP.

## 5. Conclusions

In this study, three indexes were combined with DSR, BBR, FTIR and FM testing, and AHP to comprehensively analyze various technical properties of RPP/SBS composite modified asphalt. The optimal content of RPP in the modified asphalt and its modification mechanism were quantitatively determined and compared with SBS-modified asphalt. The key conclusions of the analysis are described in the following. 

(1)The addition of RPP will significantly improve the high-temperature performance and rutting resistance of asphalt, thus reducing the temperature sensitivity of modified asphalt and enhancing its temperature sensitivity, yet it has a limited impact on the low-temperature performance of asphalt. Following the addition of SBS, the low-temperature performance of modified asphalt significantly improves. At the same temperature, the rutting factor of RPP/SBS composite-modified asphalt is about 100% higher than that of SBS-modified asphalt. According to Superpave regulations, RPP/SBS composite modified asphalt reduces the low-temperature qualified temperature of base asphalt from −12 °C to below −24 °C and increases the high-temperature failure temperature to above 76 °C, Great improvements are observed in the technical indicators of RPP/SBS composite modified asphalt, and the actual application scope has expanded.(2)The infrared spectra show that the addition of RPP does not produce a new absorption peak, and thus there is no chemical reaction between just RPP and asphalt. According to the fluorescence microscope images, RPP and SBS form a network structure in asphalt to cross-link with asphalt, which improves the stability of the asphalt binder. However, the excessive RPP addition will lead to aggregation between polymers and destroy the original structure. As a result, the storage stability and low-temperature crack resistance of modified asphalt further deteriorate. Therefore, attention should be paid to the control of the RPP dosage in practical applications.(3)The weight factor optimization system of modified asphalt was established with the analytic hierarchy process, and combined with the environmental and climatic characteristics of Northeast China, the comprehensive performance of 2% RPP/4% SBS composite modified asphalt was quantitatively optimized. Compared with the SBS-modified asphalt, the comprehensive performance of 2% RPP/4% SBS composite-modified asphalt was also significantly improved. The scores obtained from the high and low-temperature properties (amongst other properties) suggest that the modified asphalt should be used in cold and moderate-temperate regions, such as Northeast China. This method aids road builders in decision-making.

Based on the test results and analysis of the current study, compared with SBS-modified asphalt, adding the appropriate amount of RPP improves the road performance of asphalt and expands the application scope of modified asphalt in road engineering. The establishment of the optimization system provides a method for the selection of material consumption in the future. The combined micro- and macro-analysis of the modified asphalt also improve the accuracy of the test conclusions. This work is of significance for the application of waste plastics in road engineering, and the recycling of waste plastics will promote the construction of a resource-saving and environment-friendly society. Future research will focus on the optimal amount of modified asphalt in the mixture, road performance, and economic value brought by the recycling of RPP so as to more intuitively reflect its social and economic values.

## Figures and Tables

**Figure 1 materials-15-08616-f001:**
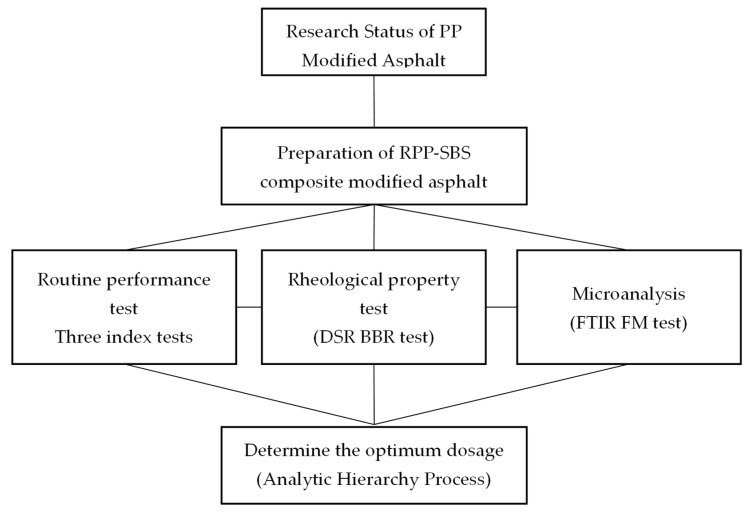
The design process of this study.

**Figure 2 materials-15-08616-f002:**
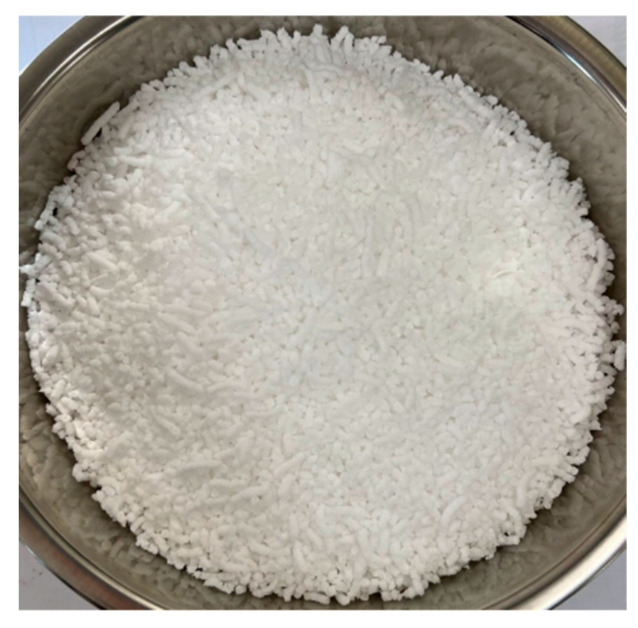
SBSYH-792E modifier.

**Figure 3 materials-15-08616-f003:**
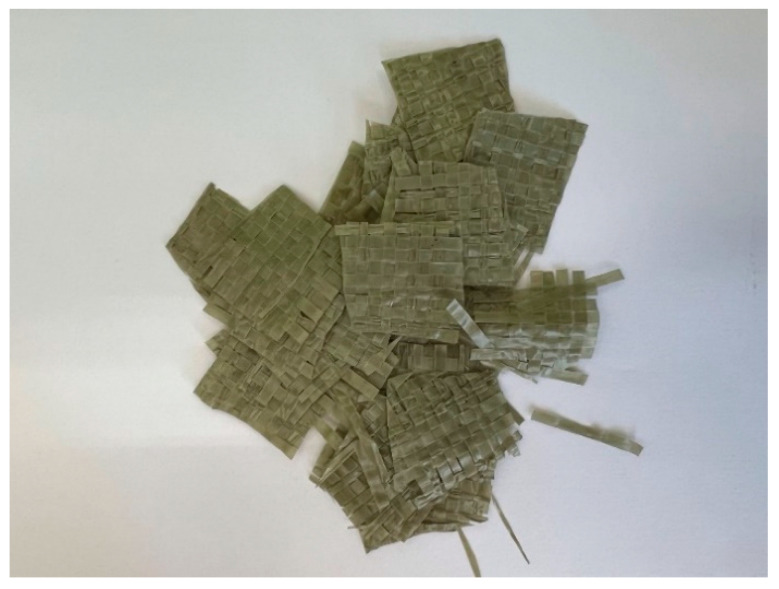
Waste PP modifier.

**Figure 4 materials-15-08616-f004:**
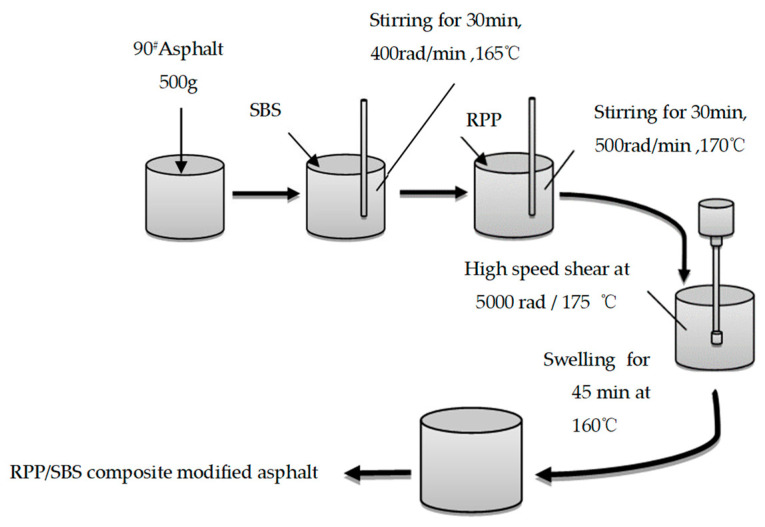
RPP/SBS composite modified asphalt preparation flow chart.

**Figure 5 materials-15-08616-f005:**
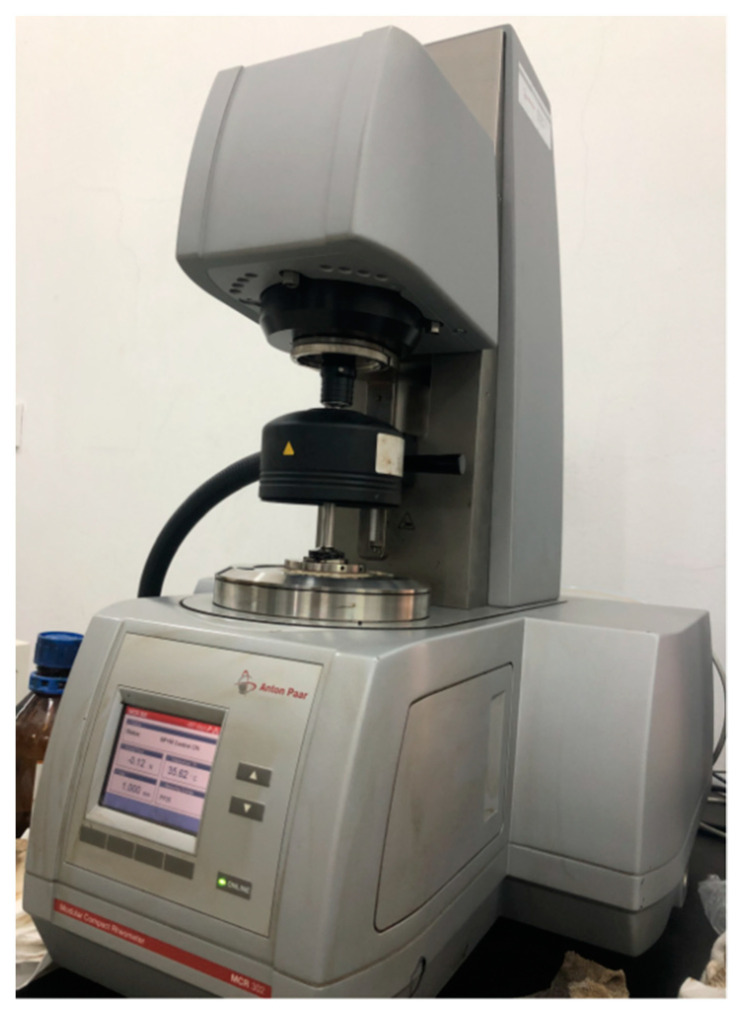
Dynamic shear rheometer.

**Figure 6 materials-15-08616-f006:**
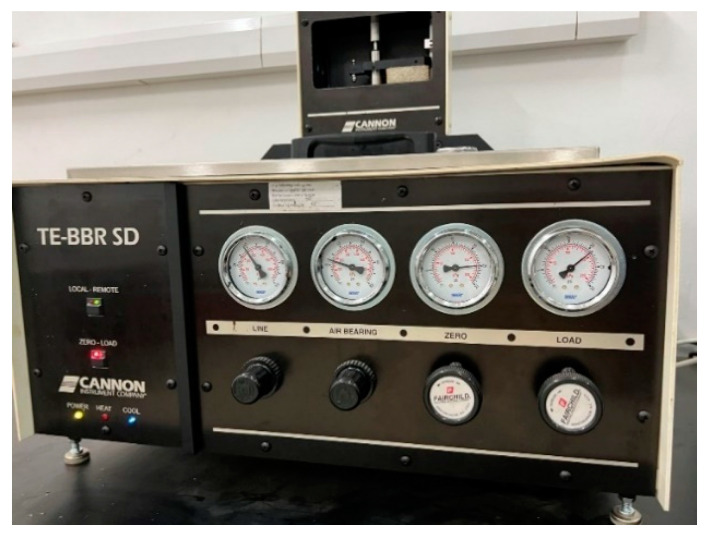
Bending beam rheometer TE-BBR-F.

**Figure 7 materials-15-08616-f007:**
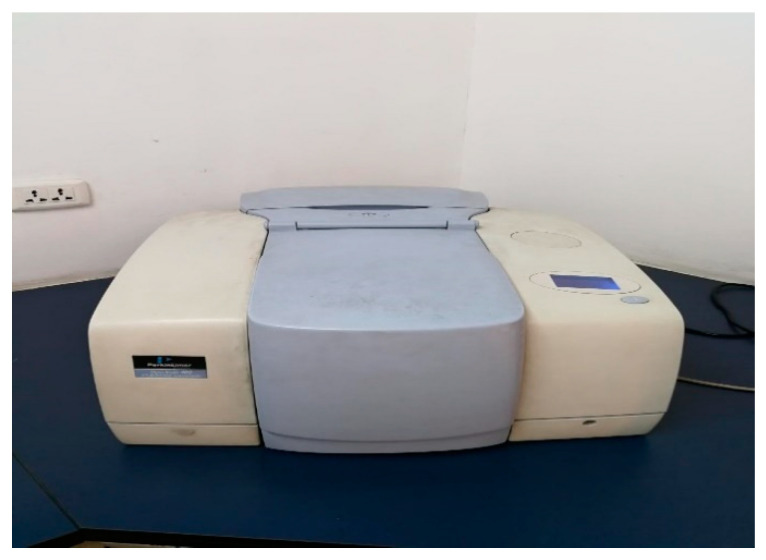
Fourier red infrared spectrum tester.

**Figure 8 materials-15-08616-f008:**
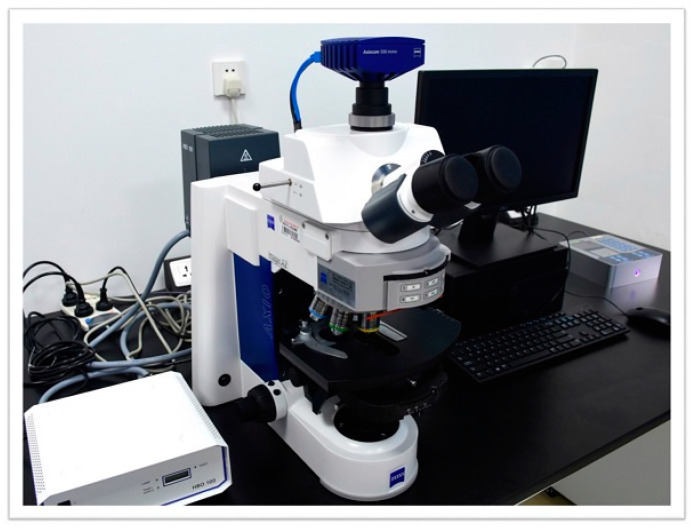
High-resolution fluorescence microscope in frontal position.

**Figure 9 materials-15-08616-f009:**
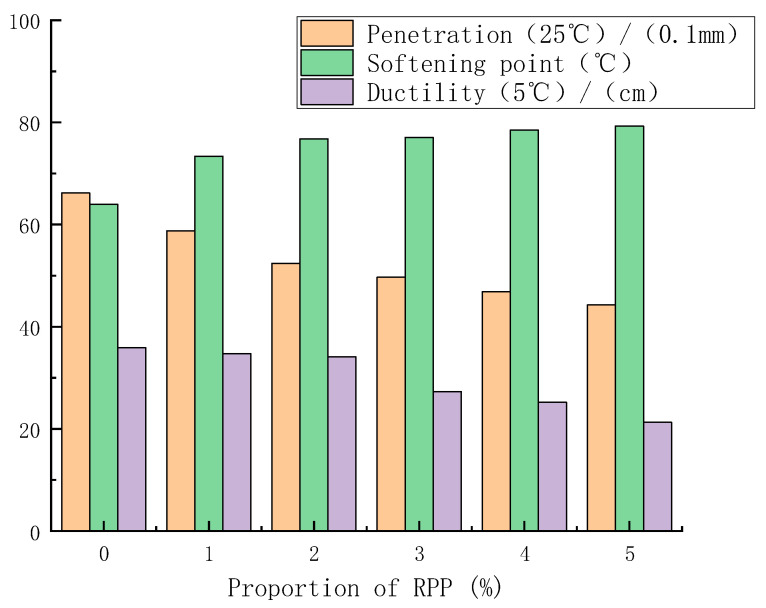
Influence of RPP content on three modified asphalt indexes.

**Figure 10 materials-15-08616-f010:**
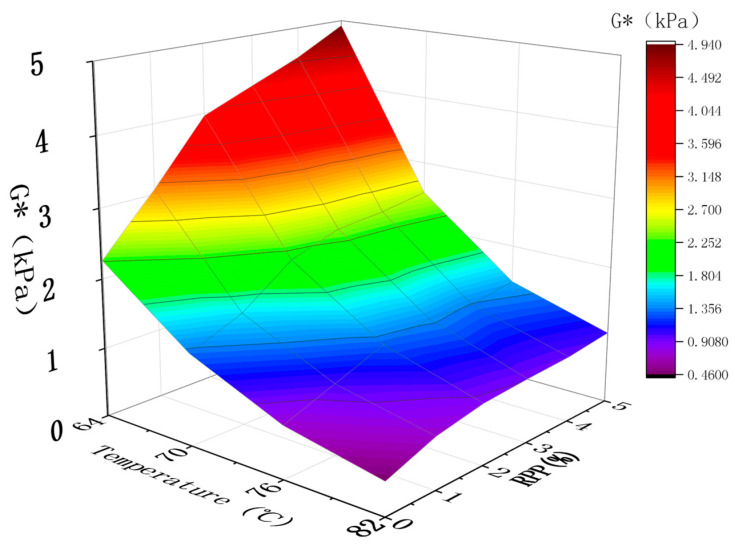
Effect of temperature and RPP content on modified asphalt G*.

**Figure 11 materials-15-08616-f011:**
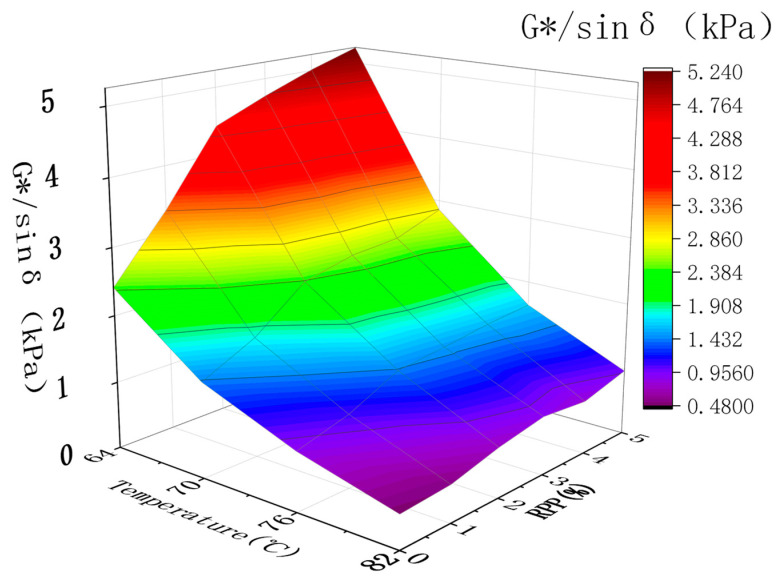
Influence of temperature and RPP content on the G*/sin δ of modified asphalt.

**Figure 12 materials-15-08616-f012:**
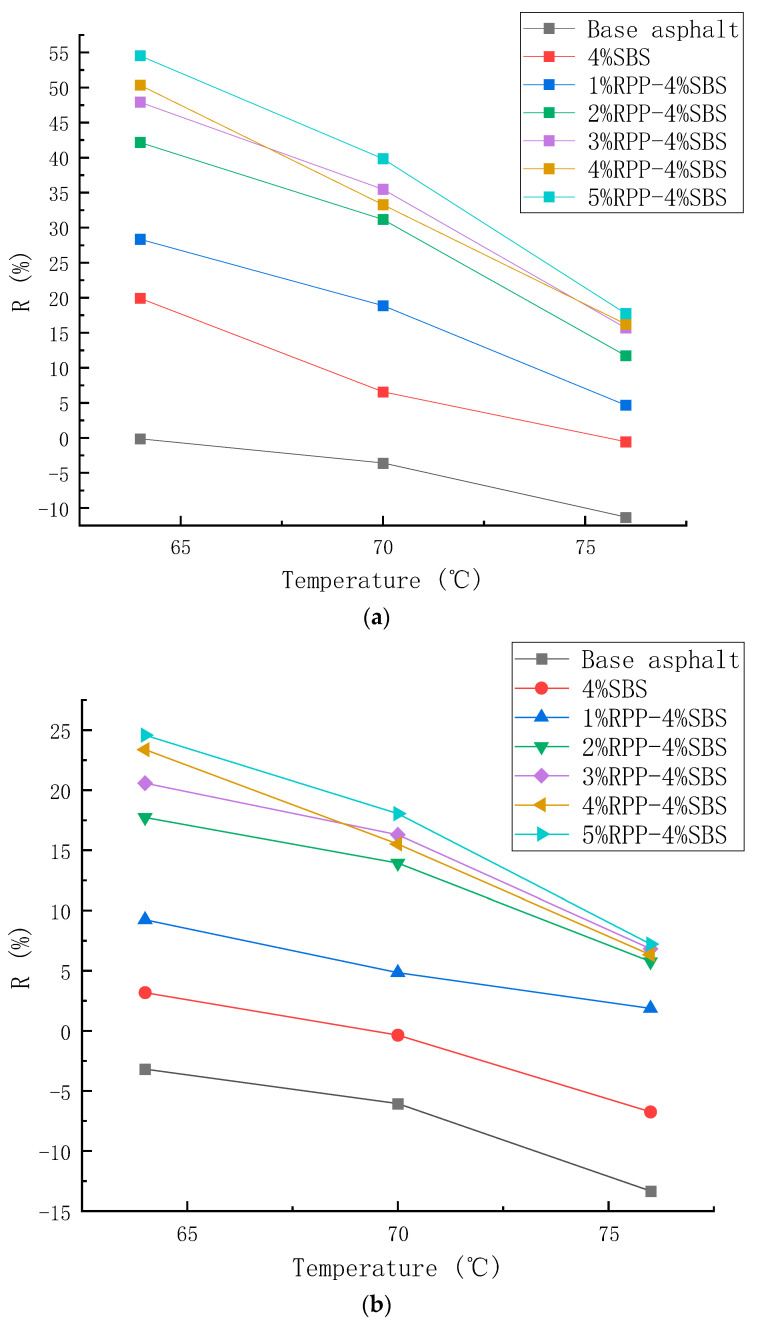
Changes in the average strain recovery rate of asphalt with temperature under different stress levels. (**a**) 0.1 kPa, (**b**) 3.2 kPa.

**Figure 13 materials-15-08616-f013:**
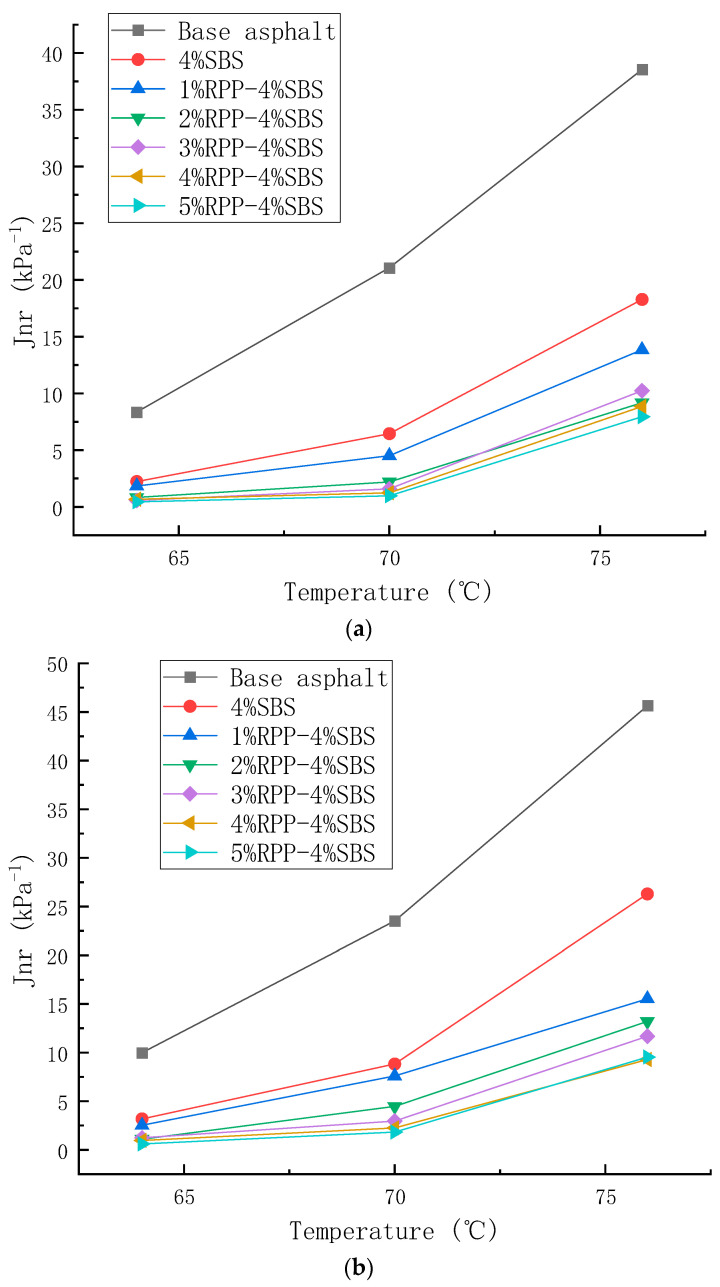
Changes in the unrecoverable creep compliance of asphalt with temperature under different stress levels. (**a**) 0.1 kPa, (**b**) 3.2 kPa.

**Figure 14 materials-15-08616-f014:**
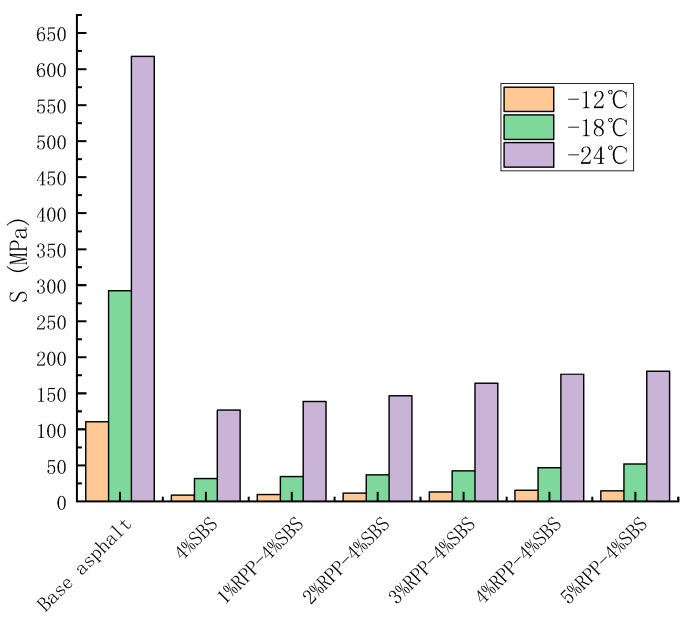
Effect of RPP content on the stiffness modulus of modified asphalt.

**Figure 15 materials-15-08616-f015:**
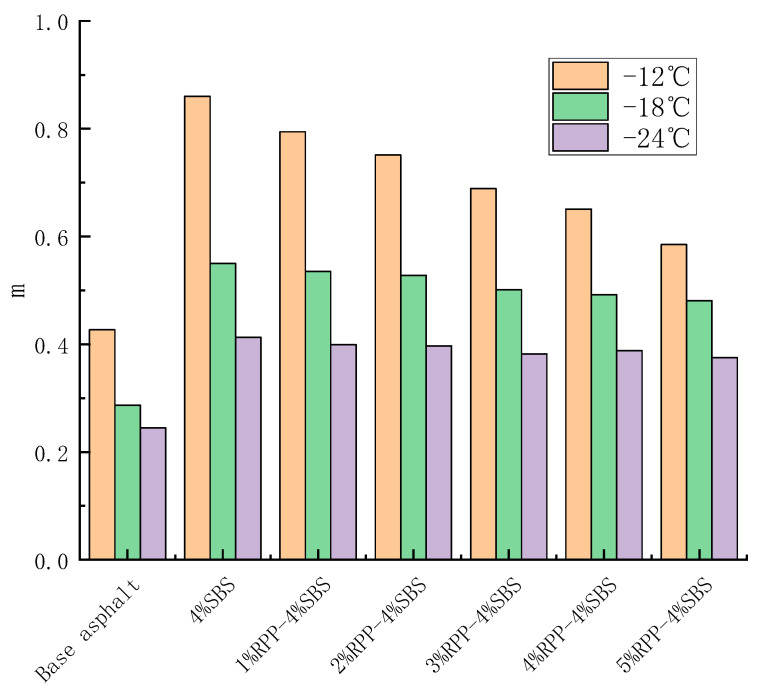
Effect of RPP content on the creep rate of modified asphalt.

**Figure 16 materials-15-08616-f016:**
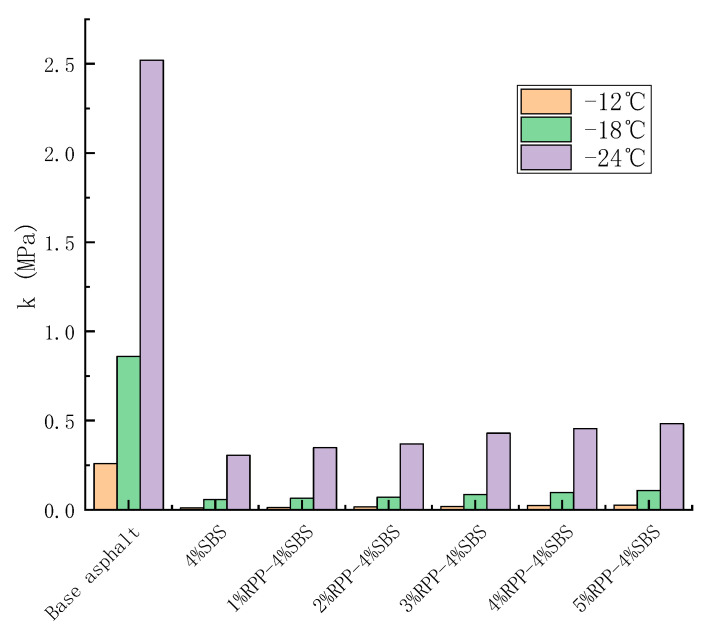
Effect of RPP content on the k index of modified asphalt.

**Figure 17 materials-15-08616-f017:**
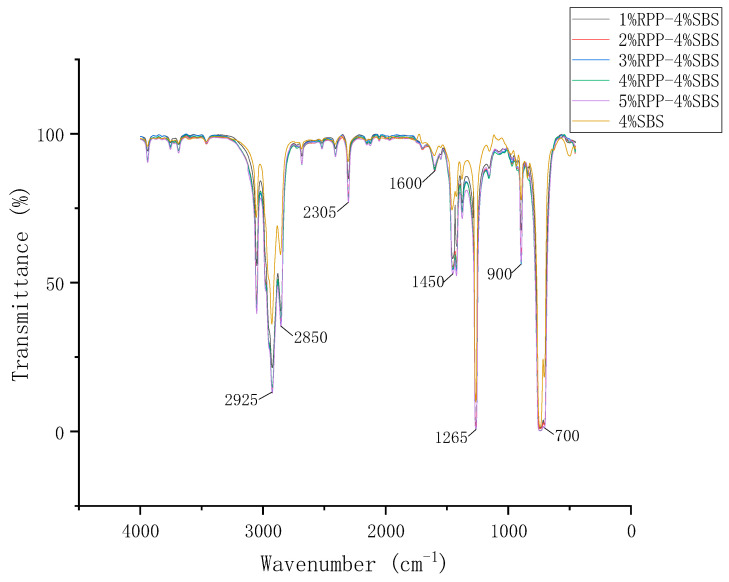
Infrared spectra of modified asphalt.

**Figure 18 materials-15-08616-f018:**
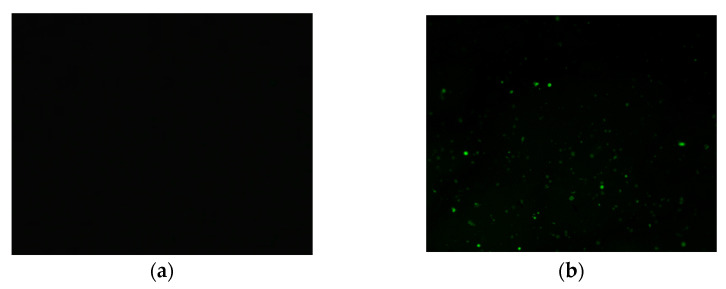
Fluorescence microscope (10 × 10) imagery. (**a**–**f**) represent matrix asphalt, 2% RPP modified asphalt, 5% RPP modified asphalt, 4% SBS modified asphalt, 2% RPP/4% SBS composite modified asphalt, and 5% RPP/4% SBS composite modified asphalt, respectively. The black area represents the asphalt phase, and the green area represents the polymer phase.

**Table 1 materials-15-08616-t001:** Basic Performance Parameters of SBS.

Project	Appearance	Molecular Structure	S/B (Mass Ratio)	Shore Hardness (A)	Tensile Strength (MPa)
Technical indicators	Linetype	White loose column	40/60	86	24.6

**Table 2 materials-15-08616-t002:** Performance parameters of SK-90 base asphalt.

Technical Index	Test Results	Technical Requirements
Penetration (25 °C)/(0.1 mm)	84.7	80–100
Softening point (ring and ball method)/°C	47.7	≥45
Ductility (5 °C)/cm	11.4	-
Dynamic viscosity (60 °C)/Pa s	178.3	≥160
Solubility (%)	102.8	≥99.9
Brookfield Viscosity (135 °C/Pa·s)	0.349	-
Density (25 °C)/(g/m^3^)	1.04	-

**Table 3 materials-15-08616-t003:** Basic Physical Properties of RPP.

Project	Melting Point (°C)	Melting Rate (g*10 min^−1^)
Test result	115–139	4.5–6.0

**Table 4 materials-15-08616-t004:** Influence of temperature and RPP content on G* (kPa) of modified asphalt.

RPP Content	64 °C	70 °C	76 °C	82 °C
0	2.260	1.310	0.744	0.470
1	3.072	1.616	1.011	0.732
2	3.963	2.143	1.279	0.867
3	4.261	2.379	1.387	0.914
4	4.554	2.435	1.345	0.947
5	4.925	2.538	1.458	1.032

**Table 5 materials-15-08616-t005:** Influence of temperature and RPP content on the G*/sin δ (kPa) of modified asphalt.

RPP Content	64 °C	70 °C	76 °C	82 °C
0	2.413	1.413	0.867	0.493
1	3.345	1.785	1.076	0.546
2	4.423	2.373	1.418	0.735
3	4.735	2.546	1.493	0.866
4	5.012	2.622	1.582	0.791
5	5.243	2.852	1.635	0.971

**Table 6 materials-15-08616-t006:** RPP content of modified asphalt.

Grade	RPP (%)
A	1
B	2
C	3
D	4
E	5

**Table 7 materials-15-08616-t007:** Summary of evaluation indexes of each modified asphalt group.

Bitumen Type	A	B	C	D	E	SBS Modified Asphalt
G*/sinδ (64 °C)/KPa	3.34	4.42	4.73	5.01	5.24	2.41
k (MPa)	0.0645	0.06996	0.08457	0.09549	0.10837	0.05766
Ductility (5 °C)/cm	33.9	31.7	27.3	25.2	21.3	38
Penetration index PI	0.94	1.26	1.07	0.84	0.73	0.87
Residual penetration ratio (25 °C)/%	69.5	77.8	81.1	72.6	66.3	73

**Table 8 materials-15-08616-t008:** Determination of adjacent grade scores.

Mark	0	6	7	8	9
G*/sinδ (64 °C)/KPa	<2	2–3	3–4	4–5	>5
k (MPa)	>1.1	1–1.2	0.8–1	0.6–0.8	<0.6
Ductility (5 °C)/cm	<20	20–25	25–30	30–35	>35
Penetration index PI	<0.7	0.7–0.8	0.8–0.9	0.9–1	>1
Residual penetration ratio (25 °C)/%	<60	60–65	65–70	70–75	>75

**Table 9 materials-15-08616-t009:** Scores for each modified asphalt group.

Bitumen Type	A	B	C	D	E	SBS Modified Asphalt
G*/sinδ (64 °C)/KPa	7	8	8	9	9	6
k (MPa)	8	8	7	7	6	9
Ductility (5 °C)/cm	8	8	7	7	6	9
Penetration index PI	8	9	9	7	6	7
Residual penetration ratio (25 °C)/%	7	9	9	8	7	8

**Table 10 materials-15-08616-t010:** Judgment matrix scale and its definition.

Importance (*r_i_* vs. *r_j_*)	Judgment Matrix Scale *r_ij_* (5/5–9/1 Scale)
Equal importance (level 0)	5/5 = 1
Slightly important (level 1)	6/4 = 1.5
More important (level 2)	7/3 = 2.33
Very important (level 3)	8/2 = 4
Absolutely important (level 4)	9/1 = 9
Intermediate state (-)	5.5/4.5 = 1.222 6.5/3.5 = 1.875 7.5/2.5 = 5 8.5/1.5 = 5.667

**Table 11 materials-15-08616-t011:** Comparison of matrix models.

Index	Elevated Temperature Property	Cryogenic Property	Temperature Sensing Performance	Ageing Resistance
Elevated temperature property	1	Importance of high temperature performance/low temperature performance	Importance of high temperature performance/temperature sensitivity	Importance of high temperature performance/aging resistance
Cryogenic property		1	Importance of low temperature performance/temperature sensitivity performance	Importance of low temperature performance/anti-aging performance
Temperature sensing performance			1	Importance of temperature sensitivity/aging resistance
Ageing resistance				1

**Table 12 materials-15-08616-t012:** Climatic element grades in Northeast China.

Climate Zoning	High Temperature Grade	Low Temperature Grade	Temperature Difference Grade	Ultraviolet Radiation Intensity Level	Coupling Grade of High Temperature and Ultraviolet Radiation Intensity
2-1 northeast region	2	1	1	2	2
2-2 northeast region	2	2	2	3	3

**Table 13 materials-15-08616-t013:** Recommended performance weights in the two climate zones.

Climate Zoning	Elevated Temperature Property	Cryogenic Property	Temperature Sensing Performance	Ageing Resistance
2-1	0.2	0.3	0.3	0.2
2-2	0.273	0.273	0.273	0.182

**Table 14 materials-15-08616-t014:** Comprehensive evaluation results of asphalt in the two climatic zones.

Climate Zoning	A	B	C	D	E	SBS Modified Asphalt
2-1	7.6	8.5	8.2	7.6	6.8	7.6
2-2	7.553	8.463	8.19	7.735	7.007	7.462

## Data Availability

Not applicable.

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
