# Peer review of "Selecting the Best Performing Modified Asphalt Based on Rheological Properties and Microscopic Analysis of RPP/SBS Modified Asphalt"

_materials, 2022, doi:10.3390/ma15238616_

Round 1

Reviewer 1 Report

this is good manuscript. there are some comments that should be done to enhance the quality of manuscript.

1- please add some relevant recent published papers.

2-add more detail about asphalt binder properties.

3- add the properties of used additives

4- how did you prepare the modified binder? based on literature? the temperature and mixing properties was based on which research?

5- it is better to enhance the discussion of results and compare your results with previous published papers.

6- perform the statistical analysis to know that the additives have significant effect or not.

7- improve the conclusion part.

Reviewer 2 Report

The paper of ((Study on Rheological Properties, Modification Mechanism and Optimization Method of Waste PP/SBS Composite Modified Asphalt)) is a good paper and can be accepted in the journal of materials after treatment the comments below.

1.   In title, please provide of using abbreviations as you can such as PP and SBS.

2.   In abstract, what is the SBS?

3.   Key words is the important words and should be repeated more than one time. However, the word “waste polypropylene” has mentioned one time in section 2.1.2 only. And micro analysis did not repeat.

4.   Please try to  insert the abbreviations of the words in a suitable table after abstract section to avoid the confuse that may occur due to similar the words with each other such as, In line 70, the sentence of ((Zhao et al. evaluated the performance of polypropylene should be “PP” not “RPP” , because RPP represent of “Recycled polypropylene”)) and so on.

5.   What is meaning of LDPE, EVA, SBSYH and others ???

6.   Please try to improve quality of figures 17.

7.   Please insert the figures 4, 5, and 6 in text and explain it in details.

Reviewer 3 Report

The manuscript entitled “Study on Rheological Properties, Modification Mechanism and Optimization Method of Waste PP/SBS Composite Modified Asphalt” presented experimental study. The influence of different parameters was studied and analyzed. The manuscript lacks clarity and needs much improvement before further processing. It seems like this paper is directly made from thesis without putting the additional efforts required for writing manuscripts.

This reviewer recommends major editing and resubmits for re-review.

Comments:

  • The English writing of the manuscript needs improvement. Therefore, it could benefit greatly from professional editing to improve technical writing and English.
  • Please mention your study limits and suggest some future research topics
  • In References, the sources are written in different styles. Please update the reference list.  It is necessary to bring in accordance with the requirements of the magazine for the design of References. If possible, indicate DOI.
  • The literature can be expanded by studying some of these papers.
    • Predicting Marshall Flow and Marshall Stability of Asphalt Pavements Using Multi Expression Programming
    •  
    • Life cycle cost analysis comparison of hot mix asphalt and reclaimed asphalt pavement: A case study
  • Please use some innovative keywords.
  • Please mention your study limits in the abstract.
  • The Conclusions should reflect what the practical application of the results obtained in this study is. In what climatic conditions should the recommendations of the authors be taken into account?
  • The authors should increase their discussion on previous related research and highlight how their study is providing a different approach or adding significantly to what has been done. The authors have to explain what is the new here in comparison with the previous studies. The novelty of the current work should be highlighted in the introduction. Please try to mention a problem that needs solving - in other words, the research question underlying your study clearer.
  • The title of the manuscript should be revised.
  • Some types of standards should be used to perform different experimental studies. Please provide details for the standards used in each study.
  • Section 4 should be discussed in detail.
  • The authors must redo the Abstract and bring it in compliance with the requirements of the journal. The scientific problem is poorly described (Background). The scientific novelty is not indicated. I recommend shortening the Abstract to 200 words. Editors strongly encourage authors to use the following style of structured abstracts, but without headings: (1) Background: Place the question addressed in a broad context and highlight the purpose of the study; (2) Methods: Briefly describe the main methods or treatments applied; (3) Results: Summarize the article's main findings; and (4) Conclusions: Indicate the main conclusions or interpretations. The abstract should be an objective representation of the article
  • It is advisable to add a flowchart at the beginning of the paper. Then the article would become more visual and structured
  • The economic aspects are also required for sustainability in social aspect. It is suggested to authors to evaluate the cost-benefit study of this as a further investigation
  • The conclusion should be an objective summary of the most important findings in response to the specific research question or hypothesis. A good conclusion states the principal topic, key arguments and counterpoint, and might suggest future research. It is important to understand the methodological robustness of your study design and report your findings accordingly. Please improve your conclusion section.

Round 2

Reviewer 3 Report

Accept